METHODS AND RESOURCES

# Individual canopy tree species maps for the National Ecological Observatory Network

**Ben G. Weinstein**[1]*, **Sergio Marconi**[1], **Alina Zare**[2], **Stephanie A. Bohlman**[3], **Aditya Singh**[4], **Sarah J. Graves**[5], **Lukas Magee**[3], **Daniel J. Johnson**[3], **Sydne Record**[6], **Vanessa E. Rubio**[7], **Nathan G. Swenson**[7], **Philip Townsend**[8], **Thomas T. Veblen**[9], **Robert A. Andrus**[9,10], **Ethan P. White**[1]

1 Department of Wildlife Ecology and Conservation, University of Florida, Gainesville, Florida, United States of America, 2 Department of Electrical and Computer Engineering, University of Florida, Gainesville, Florida, United States of America, 3 School of Forest, Fisheries, and Geomatics Sciences, University of Florida, Gainesville, Florida, United States of America, 4 Department of Agricultural & Biological Engineering, University of Florida, Gainesville, Florida, United States of America, 5 Nelson Institute for Environmental Studies, University of Wisconsin-Madison, Madison, Wisconsin, United States of America, 6 Department of Wildlife, Fisheries, and Conservation Biology, University of Maine, Orono, Maine, United States of America, 7 Department of Biological Sciences, University of Notre Dame, Notre Dame, Indiana, United States of America, 8 Department of Forest & Wildlife Ecology, University of Wisconsin-Madison, Madison, Wisconsin, United States of America, 9 Department of Geography, University of Colorado, Boulder, Colorado, United States of America, 10 School of Environment, Washington State University, Pullman, Washington, United States of America

* ben.weinstein@weecology.org

**Data Availability Statement:** The predictions, training data crops and shapefiles with predicted training crowns are available at https://zenodo.org/records/10926344. A web visualization is available

## Abstract

The ecology of forest ecosystems depends on the composition of trees. Capturing fine-grained information on individual trees at broad scales provides a unique perspective on forest ecosystems, forest restoration, and responses to disturbance. Individual tree data at wide extents promises to increase the scale of forest analysis, biogeographic research, and ecosystem monitoring without losing details on individual species composition and abundance. Computer vision using deep neural networks can convert raw sensor data into predictions of individual canopy tree species through labeled data collected by field researchers. Using over 40,000 individual tree stems as training data, we create landscape-level species predictions for over 100 million individual trees across 24 sites in the National Ecological Observatory Network (NEON). Using hierarchical multi-temporal models fine-tuned for each geographic area, we produce open-source data available as 1 km² shapefiles with individual tree species prediction, as well as crown location, crown area, and height of 81 canopy tree species. Site-specific models had an average performance of 79% accuracy covering an average of 6 species per site, ranging from 3 to 15 species per site. All predictions are openly archived and have been uploaded to Google Earth Engine to benefit the ecology community and overlay with other remote sensing assets. We outline the potential utility and limitations of these data in ecology and computer vision research, as well as strategies for improving predictions using targeted data sampling.

to preview predictions over RGB imagery: https://visualize.idtrees.org/. A csv file per site was uploaded to Google Earth engine and a public link is available as a FeatureCollection. For example, 'https://code.earthengine.google.com/?asset=users/benweinstein2010/RMNP ' is the RMNP, Rocky Mountain National Park, predictions. For more on using NEON data and earth engine, see https://www.neonscience.org/resources/learning-hub/tutorials/intro-aop-gee-image-collections. The code used in this manuscript is available both as an archive resource on zenodo (https://zenodo.org/records/10689811) and as a github repository (https://github.com/weecology/DeepTreeAttention).

**Funding:** This research was supported by the Gordon and Betty Moore Foundation's Data-Driven Discovery Initiative (GBMF4563) to EPW, by the USDA National Institute of Food and Agriculture McIntire Stennis project 1024612 and the Forest Systems Jumpstart program administered by the Florida Agricultural Experiment Station to SAB, and by the National Science Foundation (1926542) to EPW, SAB, AZ, DZW, and AS. This work was supported by the USDA National Institute of Food and Agriculture, Hatch project FLA-WEC-005944. PT acknowledges funding support from NSF Macrosystems Biology and NEON-Enabled Science (MSB-NES) award DEB 1638720 and NSF ASCEND Biology Integration Institute (BII) through DBI award 2021898. SR acknowledges funding support from NASA award 80NSSC23K0421 P00001 and Hatch project number ME022425. NGS and VER were supported by funding from NASA (80NSSC22K1625) and NSF Dimensions of Biodiversity (DEB-2124466). RAA was supported by the NWT LTER (NSF DEB-2224439), USDA NIFA McIntire Stennis project (1019284), and USDA NIFA postdoctoral award (2022-67012-37200). The funders had no role in study design, data collection and analysis, decision to publish, or preparation of the manuscript.

**Competing interests:** The authors have declared that no competing interests exist.

## Introduction

Broadscale tree taxonomic data is essential for forest management, conservation planning, ecosystem service modeling, and biodiversity research. Historically, collection of tree species data has largely relied on (1) field-censused plots ranging from dozens of individuals to several thousand trees [1] that provide high-quality data, but can only be monitored over small areas for each plot; and (2) satellite-based predictions of community-level taxonomic diversity, which can be made continuously over broad scales, but lack detailed information on individual trees [2]. Individual tree predictions from high-resolution airborne data complement these approaches by creating a bridge between high-quality, but spatially restricted, field data (e.g., [3]), and spatially continuous, but low-resolution data, from satellite or airborne sensors [4]. The spatial coverage of high-resolution airborne imagery from planes and UAVs allows a broader view of forest ecology over areas from dozens to 10,000s of hectares [5,6]. Access to these data can complement field data and global satellite monitoring to facilitate the assessment of forest structure and dynamics and how they respond to ecological processes, human management, and global change [7].

Individual tree detection is a long-standing task for remote sensing of the environment as it provides information on the densities of individual trees for large areas. Predicting the location of individual trees (e.g., [8–10]), as well delineating the extent of tree crowns (e.g., [11]), is essential in many remote-sensing workflows and has been a rich area of algorithmic research (see reviews by [12,13]). Deep learning algorithms using a combination of human-labeled imagery and field-based geospatial data have become the standard tool for tree detection for airborne RGB data [14–16]. The challenge for deep learning algorithms for tree detection is collecting sufficient training data to capture the variation in tree crown shape when applied across land-use and forest types.

After individual tree crowns have been delineated, the next step towards airborne forest inventories is to assign each crown a taxonomic label [17]. Dozens of models have been proposed using classical image processing [18], feature-based machine learning [19,20], and deep learning [21–23] but it is unclear if they are successful when applied to a variety of ecosystems with differences in tree density, abundance distributions, and spectral backgrounds. Given the very low sample sizes of training data in most studies, it is difficult to capture the range of species present and the spectral representations for each species. One proposed solution to this is using an ensemble of multiple time points of airborne imagery to improve within-site performance [24]. Sample size issues are magnified by class imbalance since the dominant taxa in many systems comprises more than 50% of training data and can thousands of times more common than the rarer species in the dataset. This imbalance makes it difficult to train large neural network models and create rigorous evaluation datasets [17].

Combining tree delineation and species classification to create broad scale tree maps is further complicated by the interaction between workflow components. Ref [25] reported that species classification decreased by more than 20% when moving from pixel-level to individual tree crown-level predictions. Changes in illumination during multiple days of remote-sensing data collection hampers generalization and species mapping has largely occurred at single flight line scales (e.g., [21]), or supported by terrestrial data in urban environments [26]. The changes in local species abundance over large areas contributes to further mismatch between training data and predicted landscapes at wide extents. Ref [27] proposed an approach to addressing these limitations by using a flexible hierarchical model structure that uses simple rules to define a series of models to create an ensemble species prediction. This approach uses both multiple views of the same crown across years, as well as a hierarchical structure to reduce the effect of species imbalance. It was effective at expanding the number of species that could

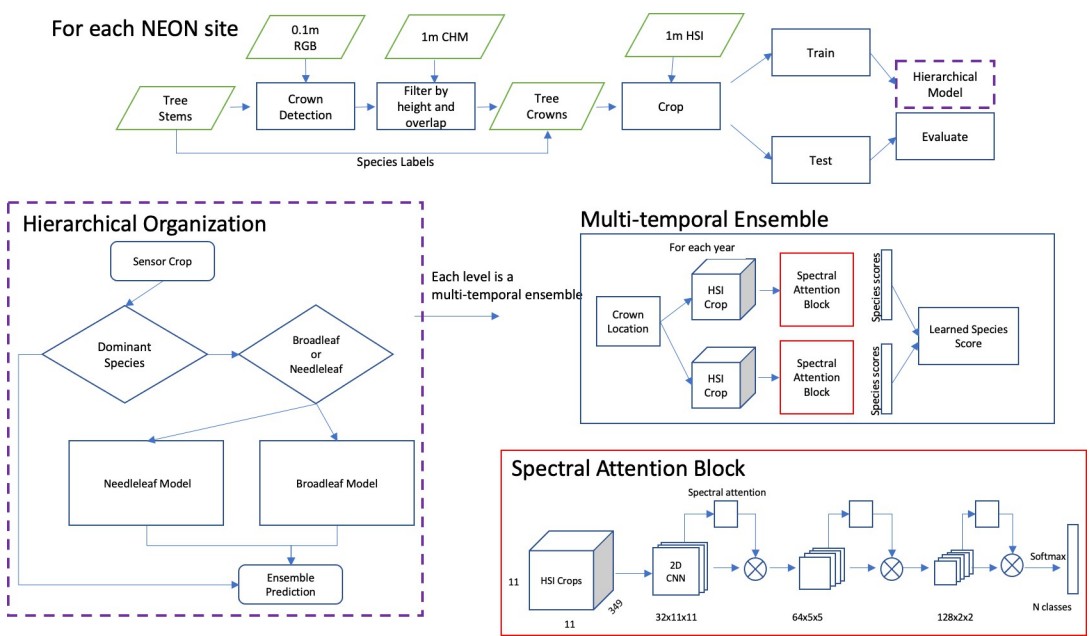

**Fig 1. Conceptual workflow for species prediction at each NEON site.** The fig is modified from [27].

be accurately classified at a single National Ecological Observatory Network (NEON) site but has yet to be tested and applied across sites with a diversity of forest types. Here, we apply the tree delineation and species classification workflow proposed for a single site in [27] to sites across the United States and assess its performance in order to provide data for ecological and computer vision research.

The NEON provides an opportunity to advance our regional scale understanding of forests by collecting open-access, high-resolution airborne remote-sensing data over 10,000s of hectares [28]. NEON collects standardized terrestrial and airborne data at dozens of sites across the US, creating an ideal situation for constructing landscape scale maps of canopy tree species for ecological research. Our aim is to generate individual canopy tree crown maps to support the ongoing forest, ecosystem, natural history, community science, and wildlife research programs at NEON sites [29–32]. Here, we combine airborne RGB, hyperspectral, and LiDAR data, to predict 100 million canopy tree locations for 81 species within 24 NEON sites across the US using machine learning models to predict crown position, species identity, health status, and height for individual trees visible in the canopy (Fig 1). Our work extends the crown location dataset published in [33] by adding predictions of species identity and alive/dead classification. The addition of species labels significantly expands the utility of this dataset for biodiversity research and natural resource management.

## Materials and methods

### Airborne sensor data

The NEON airborne observation platform (AOP) collects remote-sensing data on an annual basis during leaf-on conditions for all sites. For each site, data is collected at peak greenness to reduce variation due to phenological differences [28]. We used 4 NEON data products: (1) orthorectified camera mosaic ("RGB" NEON ID: DP3.30010.001); (2) ecosystem structure ("Canopy Height Model" NEON ID: DP3.30015.001); (3) hyperspectral surface reflectance ("HSI" NEON ID: DP1.30006.001); and (4) vegetation structure (NEON ID: DP1.10098.001).

All data were downloaded in August 2022 and were the RELEASE form [34]. The 10 cm RGB data were used to predict tree crown locations necessary for associating field labels and sensor data during model development. RGB data were also used to identify dead trees during our prediction workflow. The 1 m canopy-height model was used to determine which field collected data were likely to be visible from the air, as well as to define a 3 m minimum tree height threshold during the prediction workflow. The HSI data is used to differentiate tree species based on spectral reflectance. The HSI data spanned approximately 420 to 2,500 nm with a spectral sampling interval of 5 nm producing a total of 426 bands. NEON provides orthorectified images with a pixel size of 1 $m^2$ in 1 $km^2$ tiles that are georectified and aligned with the RGB and Canopy-Height-Model. For more information on hyperspectral data processing and calibration, see NEON technical document NEON.DOC.001288.

## Field-based species labels

The NEON Vegetation Structure dataset is a collection of tree stem points within fixed-area field plots; plot locations are allocated across sites according to a stratified random, spatially balanced design [35]. All trees in sampled areas with a stem diameter >10 cm are mapped and measured for diameter, height, health status, and species identity. Building on this NEON dataset, we contacted researchers at each NEON site to find as many mapped stems as possible outside the NEON woody vegetation sampling plots. We collected 22,072 additional canopy trees from a variety of sources, including several large ForestGEO plots co-located at NEON sites [1] and public data [36]. We followed the taxonomic hierarchy used by NEON except for genus-only, subspecies, and variety labels.

To connect species information from ground-based stem points with the airborne sensor data, we adopted a heuristic data filtering approach (Fig 2). We began with raw stem data for 41,036 individuals. We removed stems that were labeled as dead or broken, did not have a species label, or were less than 3 m in field-measured height. Whenever DBH was available, stems less than 10 cm were discarded. We then compared the field-measured height to the height of the LiDAR-derived canopy model at the stem point for the closest available year. If the difference between the LiDAR-derived and field height was more than 4 m, we discarded the stem. We then overlaid these height-filtered points to crown bounding box predictions made from the DeepForest RGB algorithm. If more than 1 height-filtered point fell within the predicted canopy crown box, we selected the tallest point using the canopy height model since this was most likely to be the dominant tree in the canopy. The shorter tree stems that overlapped the bounding box were discarded. If a point did not overlap with any bounding box, we created a 1 m buffer around the point to serve as a crown box. We refer to these crowns as "fixed boxes," and these were only included in training data, but never in testing data due to lower confidence in associating species labels and sensor pixels. Finally, if there less than 3 matched stems per species at a site, the species and its stems were removed for that site. After these steps, there were 31,736 points remaining to be used for model training and validation. Ref [20] used a portion of these training data to compare local versus global models for each site. Because of the differences in evaluation approaches, a precise comparison between [20] and this article is not possible. We emphasize that the focus of this article is on the publication of the crowns dataset rather than a comparison of a bounding box multi-temporal deep learning approach versus the pixel-based ensemble of machine learning classifiers presented in [20]

For predictions to be maximally useful, they should cover the dominant canopy tree species that occur within a site. There is a tradeoff between the filtering steps described above to strive for accurate matches with canopy trees versus a desire to include as many species as possible. We compared our final filtered data to all field-collected tree species to assess the proportion

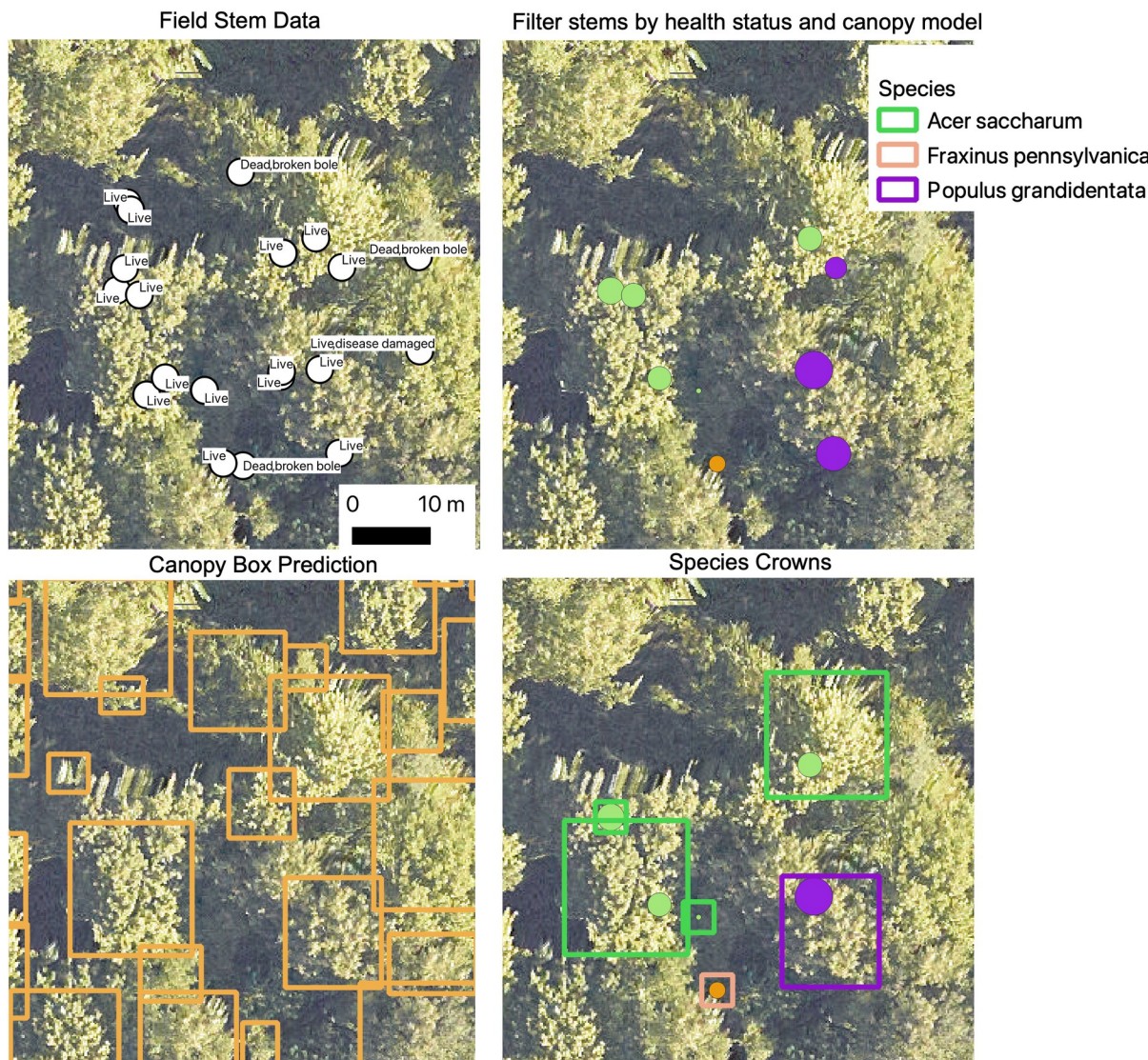

**Fig 2. Example workflow for filtering stem data to associate with crown pixel area.** Size of the dots in panels b and d are proportional to the individual tree DBH.

of field-estimated tree species richness at the site captured by the model. We calculated the proportion comparing image-predicted species data to: (1) all species—every record in the field collected data with at least 2 samples; (2) canopy species—the data filtered to 3 m height and labeled as visible in the canopy in NEON field-surveys; (3) individuals—the proportion of individuals in the training data captured by the species in the model. For example, if we had 100 individuals in a geographic site in the original field data, with 97 individuals coming from species A and 3 individuals from species B, and the model only contained species A, the proportion of species covered would be 0.5, but the proportion of individuals would be 0.97.

## Crown prediction

The DeepForest algorithm used in this work was first proposed in [37] using a combination of hand-annotated tree crown delineations and large-scale synthetic pretraining data using

LiDAR-derived tree locations; [16,38] compared the performance of tree detection algorithms across NEON sites and released the DeepForest model as an open-source python package with an average recall of 72%. Recall was measured using intersection-over-union, a common object detection metric, with a threshold for overlap of 0.4 for a positive match between predicted crown box and hand annotation. In [33], we released a dataset of 100 million crowns and calculated the performance of our workflow in matching crown predictions to individual trees by scoring the proportion of field stems that fall within a prediction. Field stems can only be applied to 1 prediction, so if 2 predictions overlap over a field stem, only one is considered a positive match. The average stem recall was 69.4%, with better performance in well-spaced western forests, and weaker performance in alpine conifer forests. DeepForest has been used widely outside of NEON sites [10,26,39,40] with accuracies generally mirroring approximately 70% for fine-tuned models from independent analysis [41].

We follow the workflow described in [33] with tree crowns less than 3 m maximum height in the LiDAR-derived canopy height model removed. Each predicted crown in the RGB imagery had a unique ID, predicted crown location, crown area, and confidence score from the DeepForest tree detection model. Following tree detection, we classified each predicted crown as "Alive" or "Dead" based on the RGB data. Presented in [27], this Alive-Dead model is a 2 class resnet-50 deep learning neural network trained on hand-annotated images from across all NEON sites. During prediction, the location of each predicted crown was cropped and passed to the Alive-Dead model for labeling as Alive (0) or Dead (1) with a confidence score for each class. Combining the information from the crown prediction, alive/dead prediction, and species classification, we release shapefiles for each 1 km NEON HSI tile that has overlapping RGB and LiDAR data (Table 1).

## Species prediction

To train species classification models, we opted to build a different model for each NEON site to create the best possible set of species predictions for downstream ecological analysis. To

**Table 1. Data available for each predicted crown.** Crowns are organized into 1 km shapefiles with UTM projection and follow the naming scheme from NEON's AOP data, with a geographic index at the top left corner. For sites with fewer than 5 species, the broadleaf and conifer labels are not available, as they are largely redundant with the species present and were all modeled jointly.

| Column name | Definition |
|---|---|
| Geometry | A 4 pointed bounding box location in UTM coordinates. |
| indiv_id | A unique crown identifier that combines the year, site, and geoindex of the NEON airborne tile (e.g., 732000_4707000). The UTM coordinate is the northwest corner of the tile. |
| sci_name | The full Latin name of predicted species aligned with NEON's taxonomic nomenclature. |
| ens_score | The confidence score of the species prediction. This score is the output of the multi-temporal model for the ensemble hierarchical model. |
| bleaf_taxa | Highest predicted category for the broadleaf model. |
| bleaf_score | The confidence score for the broadleaf taxa submodel. |
| oak_taxa | Highest predicted category for the oak model. |
| dead_label | A 2 class alive/dead classification based on the RGB data. 0 = Alive/1 = Dead. |
| dead_score | The confidence score of the Alive/Dead prediction. |
| site_id | The 4 letter code for the NEON site. See for site locations. |
| conif_taxa | Highest predicted category for the conifer model. |
| conif_score | The confidence score for the conifer taxa submodel. |
| dom_taxa | Highest predicted category for the dominant taxa mode submodel. |
| dom_score | The confidence score for the dominant taxa submodel. |

classify each predicted crown tree crown to species, we use the 1-m hyperspectral data and a multi-temporal hierarchical model. Ref [27] found that a hierarchical model outperforms a flat model by improving rare species accuracy. The hierarchical model organizes tree species into submodels, allowing each model to learn better features related to distinguishing similar classes. The submodels also allow species that are well sampled to be separated from poorly sampled species, thereby reducing the effect of class imbalance in favoring common species [42]. Within each submodel, we combine predictions for each year of available sensor data to reduce the potential overfitting and bias due to georectification of ground-truth trees and image acquisition conditions. The top model predicts "Broadleaf," "Conifer" and optionally the dominant tree species class at that site based on its frequency in the training data. A species was considered "dominant" if it consisted of more than 40% of the training samples. Without this, common machine learning approaches will predict most samples as the dominant class regardless of spectral signal. After prediction in the first subgroup, samples that are predicted as "Broadleaf" are then passed to the Broadleaf submodule, and samples that are predicted as "Conifer" are then passed to the Conifer submodule. This structure was maintained for the majority of sites, but we did allow some site-specific customization. For example, at the Ordway Swisher Biological Station, Florida (OSBS) site, the many similar oak congenerics were split off into their own oak submodule within the broadleaf submodule.

Each submodule consists of a 2D spectral attention block (Fig 1) with 3 convolutional layers and a max pooling spectral attention layer following [43]. Batch normalization is used to normalize layer weights after each convolution. This spectral attention block was repeated for each year of airborne sensor data to create an ensemble model. For example, if there are 4 years of available hyperspectral data for a geographic location, we predicted 4 classification outputs and then combined them to create the final prediction. This assumes that canopy trees at each geographic location are unlikely to change species label among years at short time scales [44]. A weighted average among all years was used to create the sample prediction for each crown. This relative weight among years was a learned parameter for each submodel. Despite multiple publications that highlight performance gains through multi-modal data fusion in remote-sensing classification [45,46] we did not find significant improvements when adding the 10 cm RGB data to species classification (Fig A in S1 File), but continue to believe it will have a role in distinguishing similar species.

For each site, we pretrained the hierarchical model using data from all sites, but only including the species at the focal site. We then fine-tuned this model using samples only at the target site. We experimented with a single NEON-wide model across all sites, but found consistently worse performance, especially for rare species (Fig B in S1 File). For each site, we pretrained for 200 epochs, decreasing the learning rate of each submodel based on performance on the focal site test data. We then fine-tuned this model with the available annotations at the target site for 200 epochs. Learning rates differed among submodules, with the dominant class and conifer submodules having an initial learning rate of 10e-5, and the broadleaf model starting at 10e-4. We allowed batch size to vary between 12 and 24 depending on the site to account for differences in class imbalance and dataset size.

To determine the evaluation accuracy of species predictions, we developed a train-test split with a minimum of 10 samples per class. To minimize the potential effect of spatial autocorrelation in hyperspectral signature between training and test datasets, we adopted a spatial block approach [17]. All samples within a NEON plot or within a 40-m grid for the non-NEON contributed data were assigned to training or test. We performed this assignment iteratively until the minimum number of samples per class were in the test dataset. The remaining samples were used to train the model. For each site, we evaluated the accuracy and precision of each species. To get the site-level score, we used both micro-averaged accuracy and macro-averaged

accuracy. Micro-averaging weights all samples the same, and therefore, is largely driven by the performance of the common species. Macro-average weights all species the same, giving greater importance to the rare species as compared to their frequency in the dataset. We also computed the accuracy of the higher order taxonomic labels (e.g., "Broadleaf" versus "Conifer"), which may be useful to downstream applications in which coarser categories are sufficient.

## Results

We developed individual canopy tree species predictions for 81 species at 24 NEON sites (Table 2). To visualize the predictions and overlapping RGB data, see visualize.idtrees.org. There was an average of 6.56 species per site, with a maximum of 15 species (Harvard Forest, Massachuesetts) and minimum of 3 (Delta Junction, Alaska and San Joaquin Experimental Range, California). Compared to reference species lists filtered for canopy species, the crown dataset covered 47.5% of the total species richness for trees ≥10 cm dbh represented in the reference list at the sites (Fig 3). These species account for an average of 85.0% of the stems ≥10 cm dbh from the forest plot data at the NEON sites. The average model had a micro-averaged accuracy of 78.8% and a macro-accuracy of 75.8% (Table 2). Sites with more data generally performed well, with a general pattern of decreasing species-level accuracy with fewer data (Fig 4). Consistent with previous work, the highest performing sites, including Teakettle Canyon, CA (TEAK), Niwot Ridge Colorado (NIWO), and Yellowstone National Park, Wyoming (YELL), were dominated by conifers and had relatively low species diversity [20]. Models performed more poorly in southern broadleaf forests, such Talladega National Forest, Alabama

**Table 2. Evaluation scores for each NEON site included in the dataset.** Sites are ranked from highest to lowest micro accuracy.

| Site, state | Forest description | Micro accuracy | Macro accuracy | Species | Train samples | Test samples |
|---|---|---|---|---|---|---|
| SJER, CA | Oak Savannah | 1.00 | 1.00 | 3 | 47 | 27 |
| GRSM, NC | Southern Hardwoods | 0.90 | 0.89 | 3 | 200 | 29 |
| TEAK, CA | Western Conifer | 0.82 | 0.83 | 7 | 713 | 67 |
| BONA, AK | Riparian and Taiga | 0.82 | 0.74 | 4 | 584 | 103 |
| STEI, MI | Northern Hardwoods | 0.80 | 0.83 | 6 | 283 | 82 |
| NIWO, CO | Alpine Conifer | 0.80 | 0.77 | 4 | 852 | 46 |
| YELL, WY | Western Conifer | 0.80 | 0.83 | 3 | 390 | 10 |
| SERC, MD | Southern Hardwood | 0.80 | 0.68 | 11 | 816 | 287 |
| DELA, AL | Southern Hardwood | 0.79 | 0.79 | 7 | 166 | 72 |
| DEJU, AK | Taiga | 0.79 | 0.78 | 3 | 571 | 52 |
| UNDE, WI | Northern Hardwood | 0.79 | 0.79 | 13 | 547 | 178 |
| SOAP, CA | Western Conifer | 0.78 | 0.78 | 4 | 223 | 37 |
| MLBS, VA | Southern Hardwood | 0.78 | 0.75 | 5 | 363 | 54 |
| TREE, MI | Northern Hardwood | 0.78 | 0.72 | 15 | 643 | 168 |
| WREF, WA | Western Conifer | 0.76 | 0.66 | 4 | 598 | 97 |
| TALL, AL | Southern Hardwood | 0.76 | 0.72 | 6 | 250 | 125 |
| HARV, MA | Northern Hardwood | 0.76 | 0.57 | 15 | 9,782 | 1,194 |
| OSBS, FL | Oak Savannah, | 0.73 | 0.63 | 14 | 3,293 | 240 |
| CLBJ, TX | Oak Savannah | 0.73 | 0.73 | 3 | 187 | 30 |
| BLAN, VA | Riparian | 0.72 | 0.73 | 8 | 271 | 79 |
| LENO, AL | Southern Hardwood | 0.71 | 0.71 | 3 | 74 | 28 |
| RMNP, CO | Alpine Conifer | 0.70 | 0.70 | 7 | 671 | 99 |
| BART, VT | Northern Hardwood | 0.68 | 0.66 | 7 | 514 | 125 |
| UKFS, KT | Southern Hardwood, Riparian | 0.60 | 0.60 | 8 | 204 | 85 |

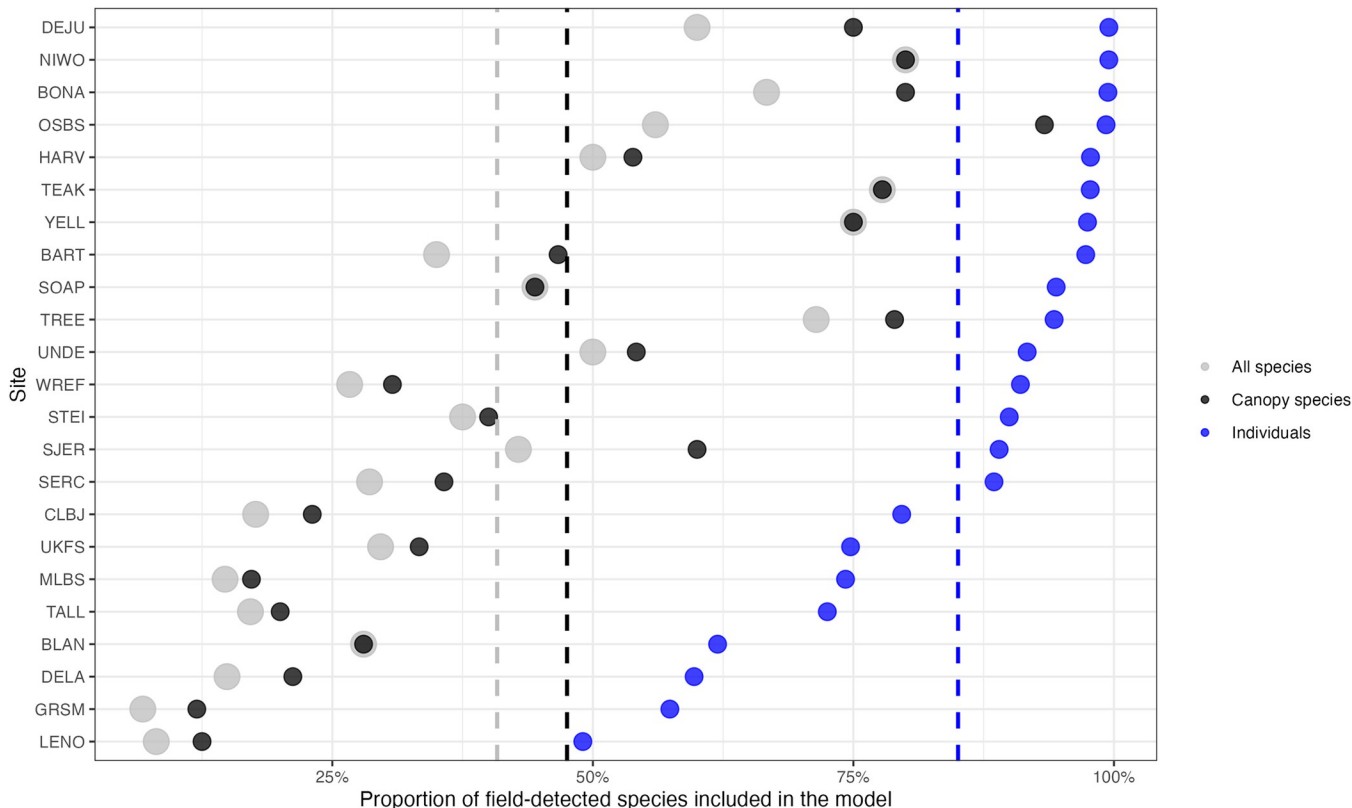

**Fig 3. The proportion of species included in the model for each site compared to species with at least 2 records in the field-collected data.** We calculated the proportion compared to: (1) all species—every record in the field collected data with at least 2 samples; 2) canopy species—the data filtered to 3 m height and labeled as visible in the canopy in NEON field-surveys; 3) individuals—the proportion of individuals in the training data captured by the species in the model. For example, the BART model has 35% of species found during field surveys, 46% of the species judged to be in the canopy, but these species represent over 97% of the sampled individuals at the site. For a complete list of each species in the model and the canopy-filtered data, see Table A in S1 File. The dashed line is the mean number of species across sites for both species and individual proportions. The underlying data for this figure can be found in supplemental data "S1 Data."

(TALL) and Smithsonian Environmental Research Center, Maryland (SERC), with higher biodiversity, closed canopy structure, and/or low data coverage per species. The most abundant species at a site typically had the highest accuracy, with lower accuracy for rarer species (Fig 4).

Applying the best model for each site to all available airborne tiles, we predicted 103,441,970 trees with an average of 4.31 million trees per site. Of the 24 sites, 17 are heavily forested with near continuous canopy cover. Sites vary in both area and forest density, with the smallest size in San Joaquin Valley, CA (SJER) with 0.85 million trees predicted, and the largest site in TreeHaven, Wisconsin with 7.1 million trees predicted. The sites with the most predicted trees tend to have high species diversity at local scales with complex, overlapping crown boundaries (Fig 5). Patterns of biodiversity are highly scale dependent with grouping of similar species in local areas and complex patterns of species patches at broader scales within the same site (Fig 6). Ranking the predicted species abundance for each site, the most predicted species represented approximately 60% of crown classifications (Fig 7). The dominant species was slightly less abundant in the southern broadleaf sites with 30% to 40% of crowns belonging to the most commonly predicted species. Viewing the predictions at the largest spatial extents, there is a broad range of species presence patterns, from sites showing highly mixed species to sites with distinct autocorrelation and species patterns at all spatial scales (Fig 8).

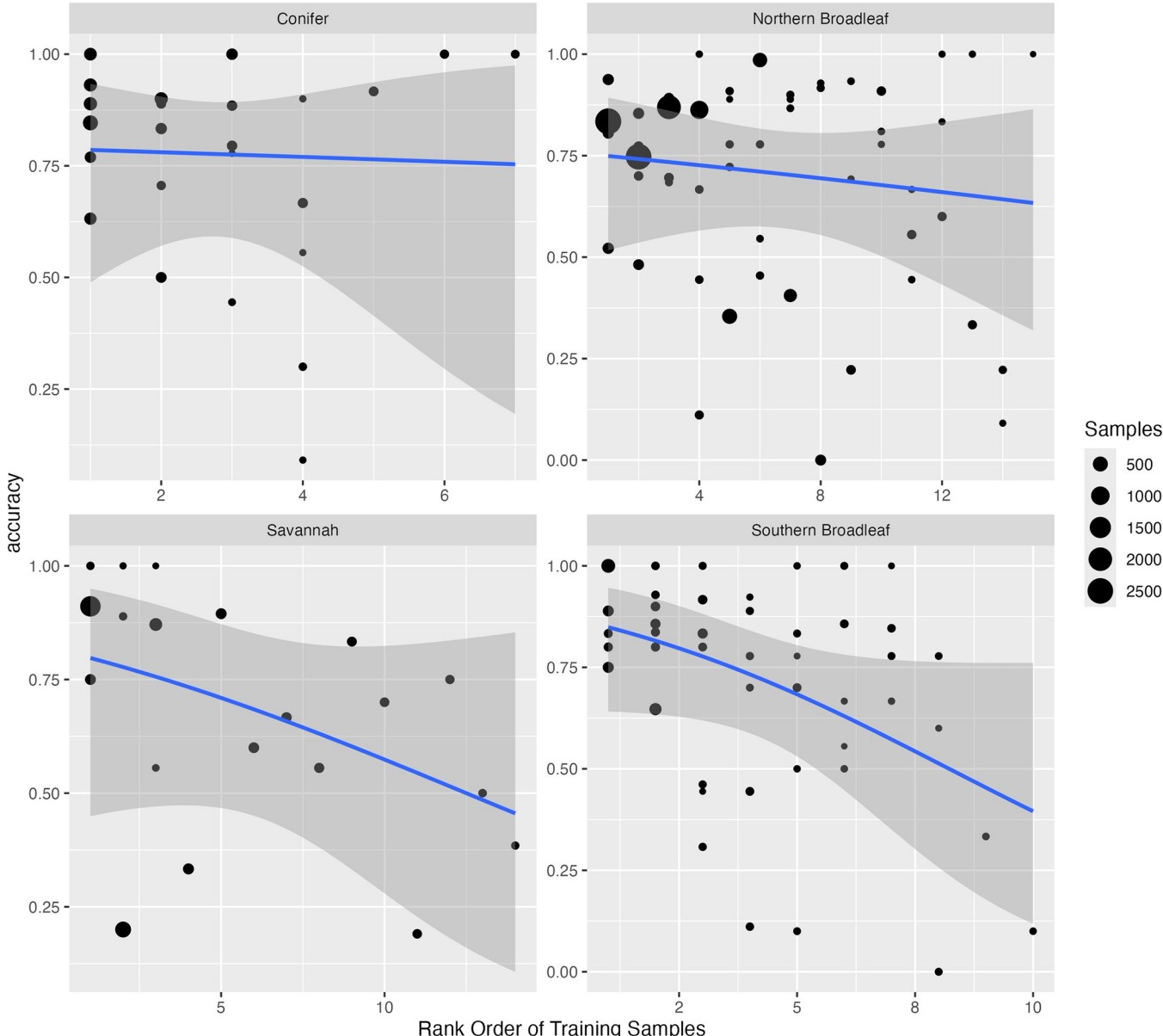

**Fig 4. Rank order abundance and evaluation accuracy for each species for each NEON site aggregated by forest type.** A binomial classification model was fit for each forest type to relate the rank order abundance of each species and evaluation accuracy. Each point is 1 species within 1 NEON site model. Point size is relative to the abundance of the species at the individual site in the training data. The underlying data for this figure can be found in supplemental data "S2 Data."

## Discussion

We used a multi-step deep learning workflow to generate individual level canopy tree species predictions continuously across large landscapes in a diverse array of forest types at sites within the NEON. The result is an extensive dataset on individual canopy tree species distribution that can be used for studying large-scale forest ecology, used as a baseline dataset for guiding field sampling, and integrated into larger scale remote sensing tasks as training data for satellite-based models. These data will inform a broad array of research programs, for example, community ecologists can study the patterns of species distributions as a function of

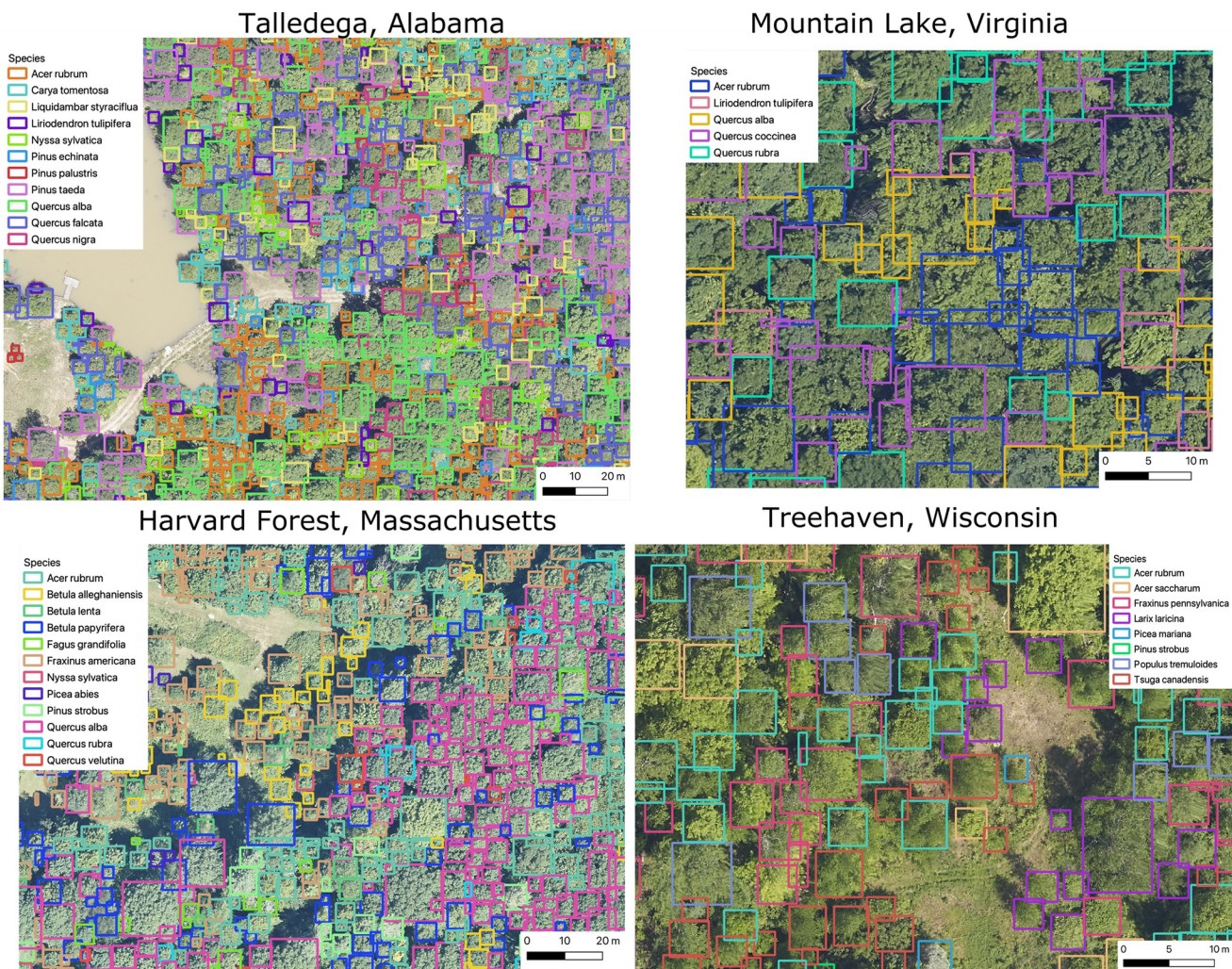

**Fig 5. Example tree detections and species labels for 4 NEON sites with closed canopy deciduous forests.**

environmental and biotic interactions [47,48], the phylogenetic structure of tree assemblages [49], and scale dependance of species plant communities [50]; ecosystem scientists can improve estimates of biomass using species-specific allometry [7,51], and foresters can measure impacts of habitat disturbance and landscape history [52,53]. To facilitate the broad use of this dataset, we have uploaded the dataset to Google Earth Engine, which provides tools and computational resources that facilitate large-scale data analysis integrating numerous remote-sensing assets that are collectively stored in the Earth Engine catalog.

The species classification models used to generate this dataset generally performed well with the accuracy for most common species ranging from 75% to 85% at well-sampled, diverse sites. Repeating a general model architecture for tree species prediction across a broad array of sites, revealed several general tendencies in the accuracy of predicted tree crowns including: (1) decreased accuracy with an increasing number of species; (2) higher accuracy at sites with more open canopy structure; and (3) a general tendency of higher performance for conifer over broadleaf species. This led to geographic patterns in accuracy even among sites in similar ecosystems, with northern broadleaf sites in general having better accuracy than the more diverse southern broadleaf sites. As local species diversity increases, classification errors are

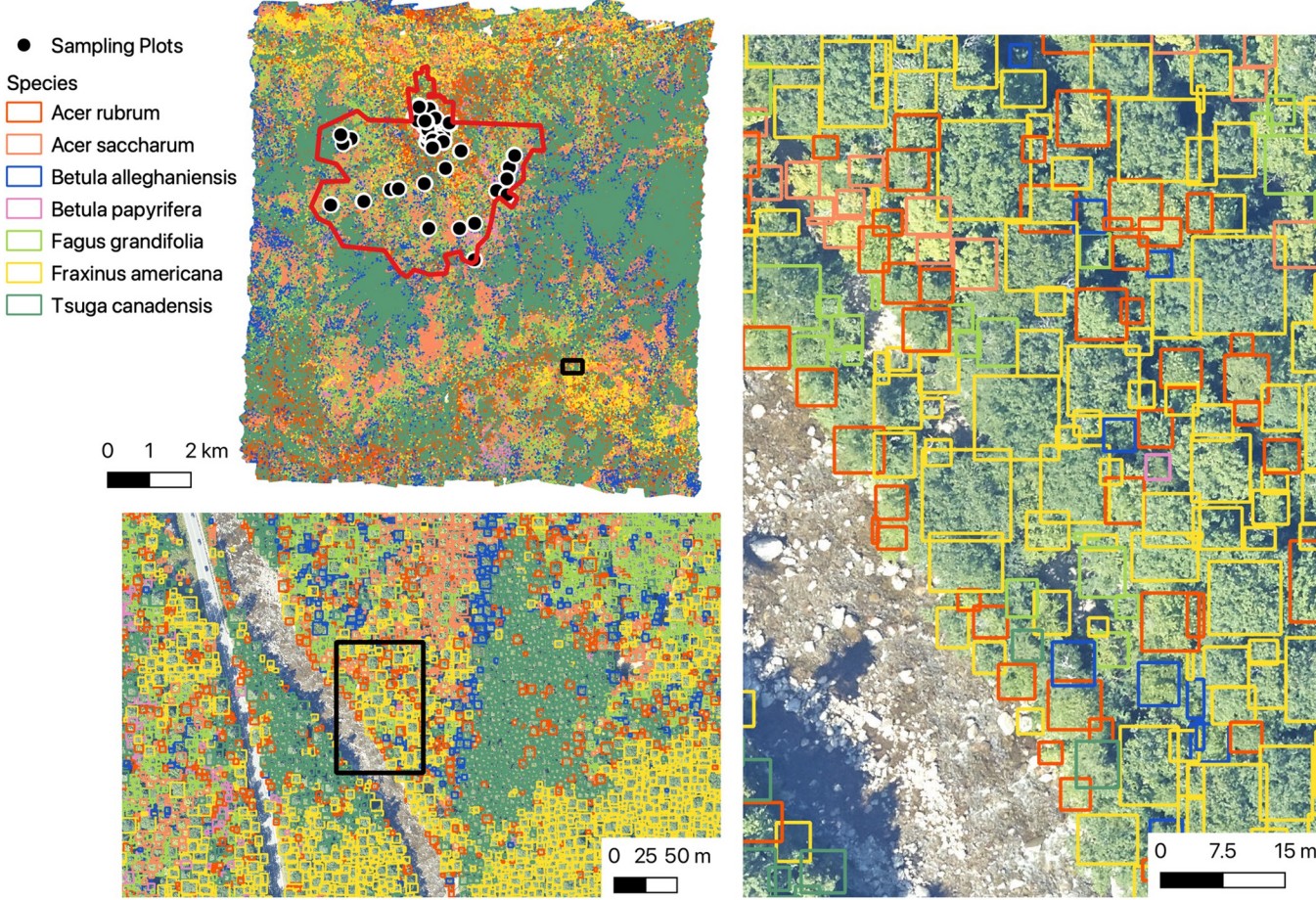

**Fig 6. Overview from Bartlett Experimental Forest, New Hampshire (BART) showing 4,352,930 tree predictions for 7 species at 3 spatial scales.** The location of NEON sampling plots and the NEON boundary are shown in the top left image.

more likely due to increased numbers of model parameters (leading to potential overfitting), greater complexity in splitting similar species, and increased frequency of neighboring trees being from different species resulting in pollution of crown edge pixels. High local turnover may also decrease accuracy because it makes training data taken from a subset of the predicted region less representative of the total biodiversity and spectral background. For example, unique habitats in the remote sensing footprint appear to be more well sampled by NEON's terrestrial plot design [35] in "Northern Broadleaf" forests than in "Southern Broadleaf" forests, likely due to the northern forests being more admixed.

Data derived from airborne remote sensing should be seen as a complement to, not a replacement for, field data. While the dataset will facilitate capturing dynamics at scales infeasible for ground-based surveys, we stress that the data are imperfect predictions that can, and should, be improved with increased data collection and model exploration. Because of the nature of the airborne data, the dataset only includes crowns in the top layer of the canopy (sunlit tree crowns), and users should be careful when calculating stand-level metrics such as abundance, crown area, or DBH and comparing them to ground-data that includes smaller subcanopy trees. Compared to field surveys, the canopy dataset will include fewer trees, with a bias towards large trees. Comparing the predicted canopy count and ground counts for the NEON field plots, the average undercount at each site was 8.51 individuals (range -2.45, 22.85)

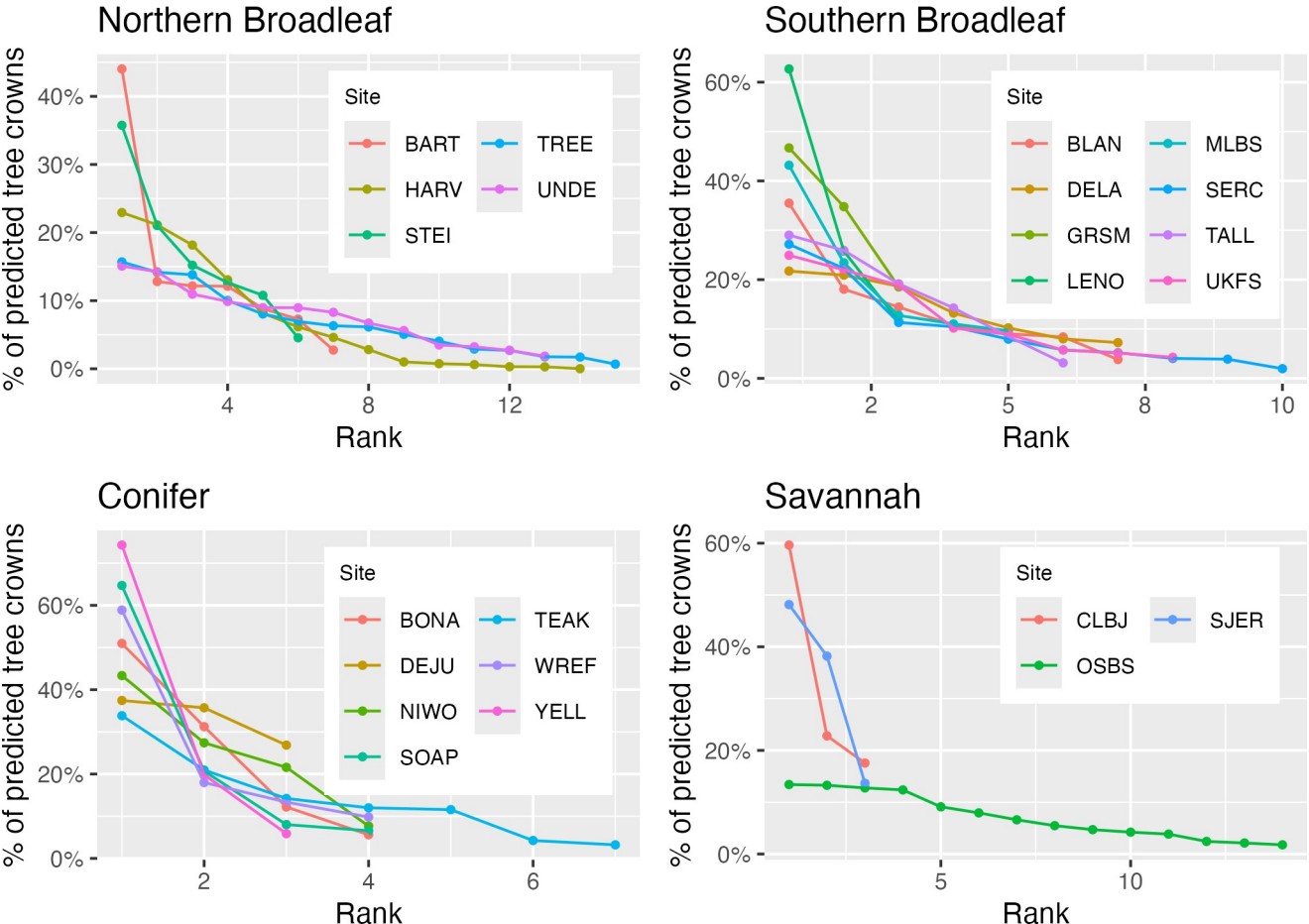

**Fig 7. Rank order abundance for the predicted crown species labels for each site.** The most commonly predicted species is rank 1, the second most commonly predicted species is rank 2, etc. Each point represents a species predicted at a site. For species identity and totals per site, see Table A in S1 File. The underlying data for this figure can be found in supplemental data "S3 Data."

(Fig C and Table C in S1 File). There will also be fewer species represented in the dataset than observed in the field, in part because subcanopy only species are explicitly excluded from the model (Fig 3).

In addition to the restriction to canopy trees, each part of the workflow has associated uncertainty and tradeoffs in defining fixed labels. DeepForest, the crown detection algorithms, has been evaluated against hand-annotated imagery [16], field-stem recall [33], and images-drawn by observers on tablets directly in the field [54], and consistently found to have roughly 70% to 75% accuracy for crown delineation. Errors occur due to over segmentation (1 tree is identified as multiple trees), under segmentation (2 or more trees are identified as a single tree), and imprecisely defined crown edges. In general, counts of canopy trees on a landscape are often more accurate (because over and under segmentation errors cancel out), but detailed boundaries and crown area are less accurate. Beyond tree detection, the alive/dead label should be interpreted as provisional since trees can lose leaves due to a variety of causes such as insect defoliation in 1 year, but ultimately recover over time [55]. Species predictions are also uncertain, and while they include the most common species at each site, they still fail to include several species that do occur in the canopy (Fig 3). The discrepancy between canopy species in the filtered field dataset and species predicted in our model is a result of several factors. Some

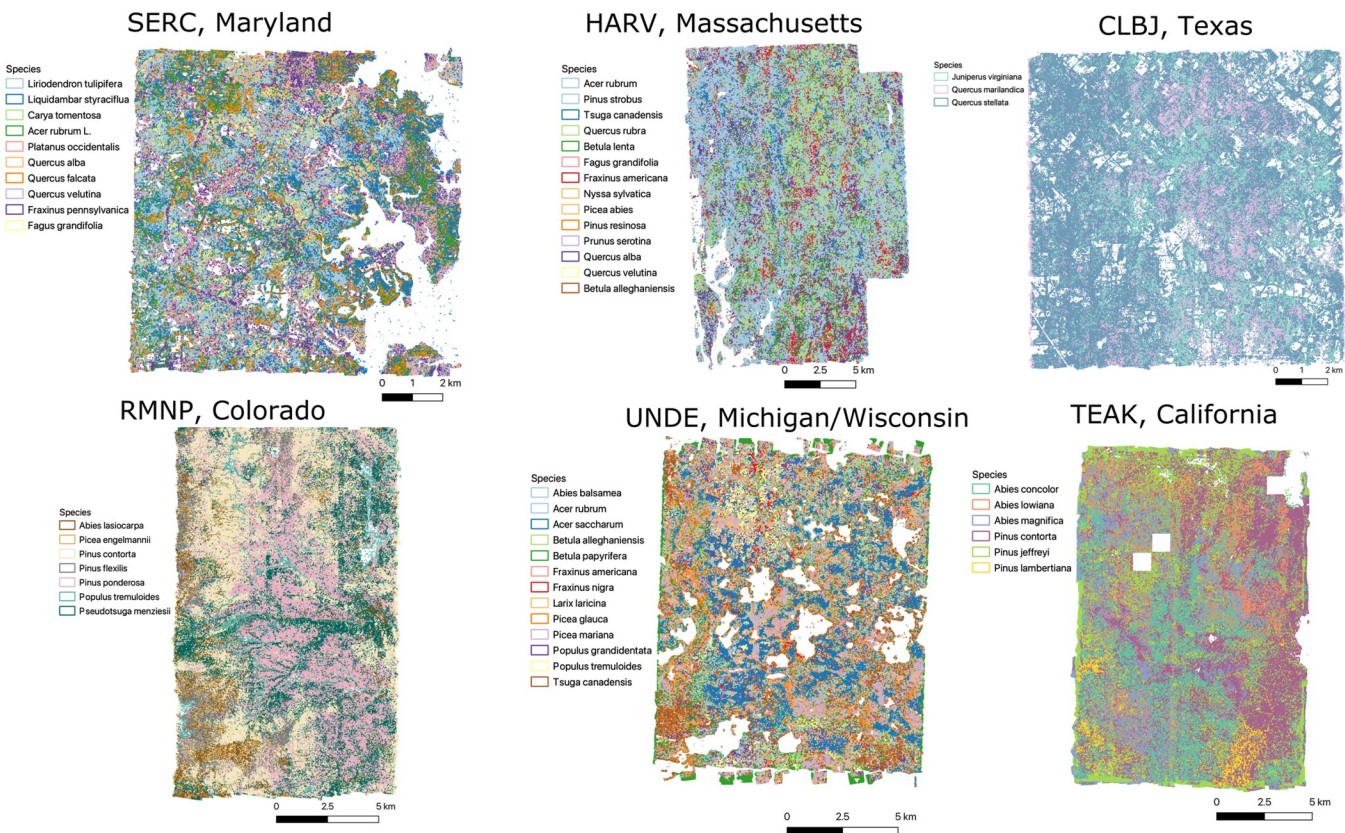

**Fig 8. Overview of multiple sites spanning a broad range of forest types.** Site names, from top left to bottom right, Smithsonian Environmental Research Center (SERC), Harvard Forest (HARV), Lyndon B. Johnson National Grassland (CLBJ), Rocky Mountain National Park (RMNP), University of Notre Dame Environmental Research Center (UNDE), Teakettle Canyon (TEAK).

canopy species are rare and thus have too few samples in our dataset to be included. This may be due to species that are common but only in rare habitats or are rare throughout a broad area of each site. On the other hand, some species may be common, but are shorter statured species that tend to mostly be in the subcanopy or only rarely reach in the canopy. When they do reach the canopy, the crowns are very small, providing poor spectral signature. Some canopy species are rare (either throughout the entire region or only occurring in rare habitats) and thus have too few samples to be modeled.

Given the uncertainties inherent in creating large-scale species maps, it is important to consider potential approaches for incorporating this uncertainty in analyses involving this and similar datasets. Ref [27] outlined multiple options for incorporating model uncertainty when using the data in downstream analysis. We compared data uncertainty through multiple training and test splits, model uncertainty by repeatedly training the model from the same training data, and prediction uncertainty using a multinomial draw of the confusion matrix to generate predicted counts for each species within a single site. While this is a useful first step, ultimately hierarchical models that can directly incorporate model uncertainty should be developed to improve downstream ecological analyses of remote sensing based data (e.g., [56]). Calibrating confidence scores using held-out data from training or test is an important step in this direction [57], but there was insufficient data to set aside for this purpose while maintaining less common species in the model. This will be a common limitation in ecological studies where the limited data can be crucial for improving model accuracy and incorporating rarer species.

Post hoc corrections of predicted counts (e.g., [58]) or models that account for multiple types of uncertainty will be crucial in making robust predictions at larger spatial extents going forward.

The process of making predictions for 100 million trees across a broad range of habitat types helped identify areas for improvement in computer vision needed to address obstacles in assembling tree maps at massive scales. The main obstacle to improving model accuracy is the availability of training data. We have found that targeted sampling can yield 10% to 20% improvements in accuracy, and significantly broaden the number of species included in the model predictions, with only a few days or weeks of field work (Box 1). The simplest form of data needed is a geospatial point of a tree stem (precise enough to ensure it falls within a predicted crown box) and its species label. Data collection should focus on less common species, since more data on common species will have limited impact on model performance. Strategies for prioritizing new data collection include: (1) using expert knowledge to identify areas containing underrepresented species; (2) using the model confusion matrix and predictions from the initial model to select species with unexpected confusion patterns, such as underrepresented species that are not visually similar that are confused by the model (a possible indicator of spectra being polluted by neighboring trees); and (3) sampling individuals with low confidence scores for their species predictions indicating either poor model performance or a species not included in the model.

There are also areas for improvement in associating tree stems with crown pixels. Our models perform better in open forests with low diversity, where spacing among trees improves crown delineation and fewer species reduces the chance of neighboring tree species polluting the spectral signature. This can be partially overcome by using crown polygons drawn on a tablet in the field, rather than relying on stem points taken by a GPS. Even a limited number of these crown polygons could allow the adoption of "weak labeling" approaches common in computer vision that rely on access to a small number of confident samples and a larger set of less confident samples.

One of the reasons additional data collections can be beneficial is that compared to the typical computer vision application, the data sample sizes of the classes used in these models are extremely low. Therefore the emerging area of research on "few shot learning," in which foundation models are used to predict new classes with only 1 to 5 samples, may be a useful avenue

## Box 1. In-depth examination of new data collection to improve models

To increase the species coverage and accuracy of these models, we need additional data collection at each NEON site. Here, we outline one effort by N.G. Swenson and V.E. Rubio to improve the model at the University of Notre Dame Environmental Research Center (UNDE) site through targeted data collection (Fig 9). The original model had 67.8% micro-accuracy, 61.6% macro-accuracy, and included 12 species. Overlaying the predictions over a recently mapped forestry plot, 3 areas of need were identified: (1) several key species were missing from current predictions; (2) there was overprediction of *Fraxinus nigra* compared to the abundance expected by field researchers; (3) there was high confusion between 2 closely related *Populus* species. Using these goals to target trees, data on an additional 157 stems of 12 species were collected along easy to access roads and forest edges. After training the model on the additional stems, the micro-averaged accuracy increased from 67.8% to 77.7% and the macro-averaged accuracy increased from 61.6% to 79.1% while adding an additional species to the test dataset.

The accuracies of the 2 closely related *Populus* species increased from 66% and 54% to 72% and 82%, respectively.

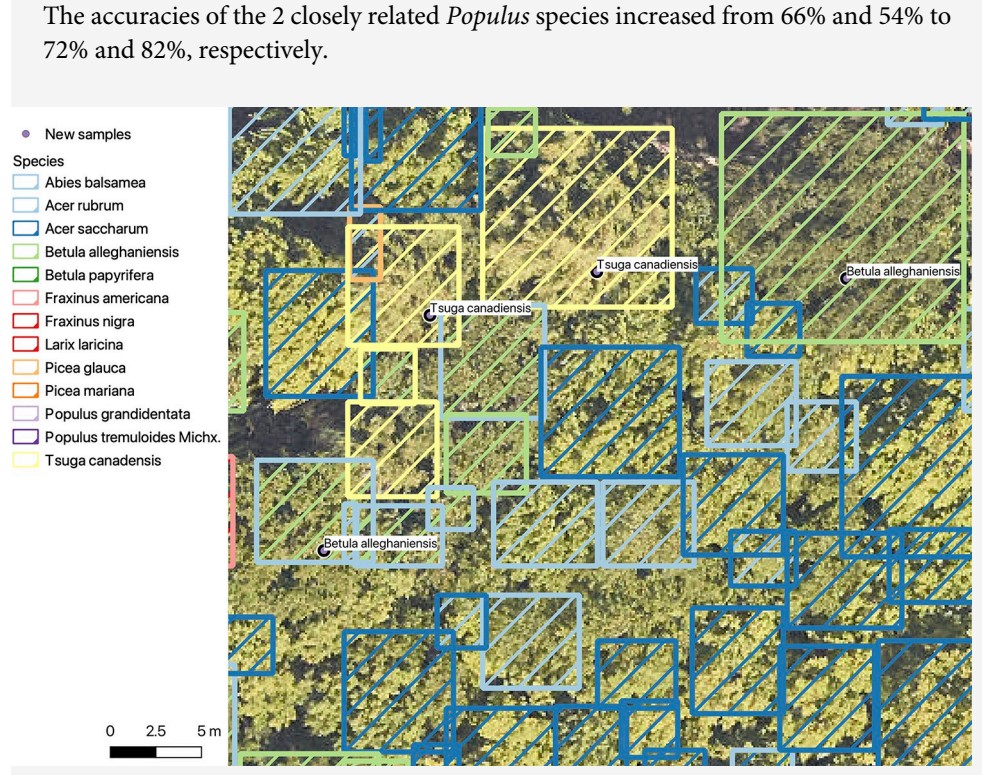

**Fig 9. Original versus re-trained model predictions for UNDE.** New sample trees were collected in the field without guidance from the predictions. The outline color is the original label, the filled shade is the revised label. The 2 *Tsuga canadensis* (top center) and the field samples were correctly predicted in the original model. The *Betula allenghensis* field samples were split. The tree on the right was correctly predicted in both models. The tree on the left was originally predicted as *Acer rubrum* but was correctly predicted in the revised model. Overall, most labels do not change among models, with only a small number of trees changing labels. For example, several trees that were originally predicted as *Acer rubrum* have been revised, and a single *Picea glauca* was revised to *A. rubrum* in the top left.

for further improving tree species predictions (e.g., [59]). In the extreme, the task of zero-shot learning [59,60], or unknown class detection, in which the model can identify classes not included in training, will help address the challenge of identifying individuals not included in the models and have utility in rapid applying models trained on NEON data to new areas. This approach is limited by our current modeling design since the site-level model approach limits portability, and the hierarchical organization can be cumbersome to apply in new regions and as new species are added. While we chose this approach because it currently produces the most accurate predictions and therefore the best resulting dataset, a single NEON-wide model that is robust to class imbalance, but maintains good separability among co-occurring species, would be a major step forward.

Extending the models used in our workflow to non-NEON sites will be important in broadening access to high-quality tree species prediction. There is considerable interest in developing species predictions for large areas using high-resolution satellites and UAVs with low-cost hyperspectral sensors. Using NEON data as a source for training data to project into these coarser resolution data has large benefits since the NEON data is both high spectral and spatial resolution. This kind of "Domain adaptation" is an open challenge in computer vision, with

many proposed approaches to try to align either the input data or learned features among disparate sensors or geographic areas [61]. The ample unlabeled airborne data at NEON opens the possibility of a combination of supervised and unsupervised learning to increase transferability among geographic sites, spectral resolutions, and spatial scales. In conjunction with automated methods for data collection, these approaches will move the community towards airborne classification models for tree species that can generalize across sampling events, geography, and acquisition hardware.

As the number of researchers working at NEON sites increases, the diversity of overlapping datasets will foster richer areas of understanding for forest ecology and ecosystem functioning. The goal of this work was to provide initial predictions for canopy trees at the landscape scale to document the broad pattern of tree species distributions, which in turn influence ecological communities and nutrient cycling. Combining these data with organismal surveys, fine-scaled environmental data, and landscape history will bring greater insights into the mechanisms underlying forest distribution and function. NEON's on-going data collection will allow these maps to be updated both in terms of geographic coverage, as well as temporal change in species abundance and individual traits.

## Supporting information

**S1 Data. The underlying data for Fig 3.**
(CSV)

**S2 Data. The underlying data for Fig 4.**
(CSV)

**S3 Data. The underlying data for Fig 7.**
(CSV)

**S4 Data. The underlying data for Fig B in S1 File.**
(CSV)

**S5 Data. The underlying data for Fig C in S1 File.**
(CSV)

**S1 File. Supplemental materials.** Table A. Species included in each model for each NEON site. The number of samples (*n*) for each species in the canopy filtered data. To be included in the model, a species needs to have at least 10 training samples and 10 test samples at a site in the final filtered data. The number of predicted trees at each site, the proportion of total predictions at the site, and the rank abundance of each species is shown. Fig A. An example model architecture for data fusion between 1 m HSI data and 10 cm RGB for tree species classification. In this example, a batch of crowns (*n* = 20), each with an HSI and RGB pair, is run through the network to jointly predict tree classes (*n* = 10). The RGB model was a resnet-50 pretrained backbone, a common RGB architecture for image-classification. The HSI architecture was the same spectral attention network used throughout the rest of the paper. The 2 features were min-max normalized separately before combined and a joint classifier was used to predict tree species classes. Table B. Experiments comparing RGB, HSI, and joint model for a single NEON site (OSBS). The experiments were done without the hierarchical model or multi-temporal ensemble approaches to highlight the difference solely from source data type. Fig B. Comparison of site-level performance for modeling workflows that use training data solely from a single site ("per-site") and pool training data across all sites "NEON-wide." Micro averaged recall is the proportion of correctly predicted ground truth stems. Macro-averaged recall is the average recall per species, thereby weighing all species equally regardless of

abundance. Several sites (JERC, MOAB, SCBI) lacked site-level predictions because the sample size per species at the individual site was too low. For the underlying data, see S4 Data. Fig C. Predicted canopy trees versus the count of all field measured trees in the NEON Woody Vegetation Structure plots. For each NEON site, the number of tree detections in the prediction data is compared to the number of field-measured detections for that NEON subplot. For the underlying data, see S5 Data. Table C. Mean differences between predicted and observed counts, and RMSE for a generalized linear model with Poisson link function between field-measured counts of all trees and predicted canopy tree count (Fig C in S1 File).
(DOCX)

## Acknowledgments

We would like to thank NEON staff and in particular Tristan Goulden and Courtney Meier for their assistance and support. We thank Natalie Heaton, Nicollete Lyons, Matthew Raulerson, Alex Seeley, Camille Sicangco, Luis Tirado, and Stuart Wilkin and for field data collection efforts.

## Author Contributions

**Conceptualization:** Ben G. Weinstein, Sergio Marconi, Alina Zare, Stephanie A. Bohlman, Aditya Singh, Sarah J. Graves, Lukas Magee, Daniel J. Johnson, Nathan G. Swenson, Philip Townsend, Thomas T. Veblen, Ethan P. White.

**Data curation:** Ben G. Weinstein, Sergio Marconi, Stephanie A. Bohlman, Sarah J. Graves, Lukas Magee, Daniel J. Johnson, Sydne Record, Vanessa E. Rubio, Nathan G. Swenson, Philip Townsend, Thomas T. Veblen, Robert A. Andrus, Ethan P. White.

**Formal analysis:** Ben G. Weinstein.

**Funding acquisition:** Ben G. Weinstein, Alina Zare.

**Investigation:** Ben G. Weinstein, Alina Zare, Aditya Singh, Lukas Magee, Ethan P. White.

**Methodology:** Ben G. Weinstein, Stephanie A. Bohlman, Aditya Singh, Sarah J. Graves, Ethan P. White.

**Project administration:** Ben G. Weinstein.

**Resources:** Ben G. Weinstein.

**Software:** Ben G. Weinstein.

**Validation:** Ben G. Weinstein.

**Writing – original draft:** Ben G. Weinstein, Sergio Marconi.

**Writing – review & editing:** Ben G. Weinstein.

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
