## [Editor Report · Decision Letter 0]

13 Nov 2023

Dear Dr Weinstein, 

Thank you for submitting your manuscript entitled "Individual tree species maps for the National Ecological Observatory Network" for consideration as a Methods and Resources paper by PLOS Biology.

Your manuscript has now been evaluated by the PLOS Biology editorial staff, as well as by an academic editor with relevant expertise, and I'm writing to let you know that we would like to send your submission out for external peer review.

Once your full submission is complete, your paper will undergo a series of checks in preparation for peer review. After your manuscript has passed the checks it will be sent out for review. To provide the metadata for your submission, please Login to Editorial Manager (https://www.editorialmanager.com/pbiology) within two working days, i.e. by Nov 15 2023 11:59PM.

Kind regards,

Roli Roberts

Roland Roberts, PhD

Senior Editor

PLOS Biology

rroberts@plos.org

---

## [Decision Letter · Decision Letter 1]

24 Jan 2024

Dear Dr Weinstein,

Thank you for your patience while your manuscript "Individual tree species maps for the National Ecological Observatory Network" was peer-reviewed at PLOS Biology. It has now been evaluated by the PLOS Biology editors, an Academic Editor with relevant expertise, and by three independent reviewers. 

You'll see that reviewer #1 says that your study is important and “of great merit,” but wants re-structuring of the Intro (including better coverage of the existing literature) and some clarification of the methodology. Reviewer #2 is also positive, but wonders if your could test the robustness of several aspects of your approach. Reviewer #3 is more critical; s/he appreciates the scale but is less impressed by the quality, urging you to flag the limitations more prominently so that readers aren’t misled into thinking that the dataset is better than it is. As part of this, s/he wants them to quantify the biases inherent in their dataset. Like reviewer #1, reviewer #3 thinks that much of the Intro belongs in the M&M, and that instead the Intro should cover the current state of the field in more detail; the Discussion should contain an account of the limitations. And you also need to provide the code (we would insist on this anyway).

In light of the reviews, which you will find at the end of this email, we would like to invite you to revise the work to thoroughly address the reviewers' reports.

Given the extent of revision needed, we cannot make a decision about publication until we have seen the revised manuscript and your response to the reviewers' comments. Your revised manuscript is likely to be sent for further evaluation by all or a subset of the reviewers.

**IMPORTANT - SUBMITTING YOUR REVISION**

*Re-submission Checklist*

*Published Peer Review*

*PLOS Data Policy*

*Blot and Gel Data Policy*

Sincerely,

Roli Roberts

Roland Roberts, PhD

Senior Editor

PLOS Biology

rroberts@plos.org

REVIEWERS' COMMENTS:

Reviewer #1:

[identifies himself as Aland H. Y. Chan]

General comments

The paper describes a database of identified and species-classified tree crowns across 24 National Ecology Observation Network (NEON) sites in the US. The distribution of tree species and how it relates to both climatic, geological, and other environmental variables is an important area of study. The database, if made available publicly, would be of great merit to the ecological community. The manuscript is well-written and reads well but would benefit from a restructuring of the introduction and some clarification over several steps in the methods. I would recommend publication after some corrections. 

Major issues

The current four paragraphs of the introduction are structured as follows: 

(1) Individual tree crown (ITC) based dataset on tree species distribution is valuable in ecology, forest management, and global change biology

(2) NEON provides a high-resolution remote-sensing dataset covering 10000s of hectares of forests

(3) The NEON dataset was used to identify 100 million trees using a modified method from Weinstein et al. (2023)

(4) The structure of the dataset and how the methods improved on existing deep learning workflows

Paragraphs (2), (3), and (4) contain a lot of information on how this study was conducted, which should largely be moved to the methods section. On the other hand, the introduction has yet to provide a proper account of existing work in the field. There is a large body of research that attempted individual tree crown species classification using different types of remote sensing data. For instance, Fassnacht et al. (2016) reviewed >100 studies that used hyperspectral, multispectral, LiDAR, or a combination of RS datasets to classify trees into species. The authors claimed that the existing studies have not applied the models in large geographical areas across biomes (L90-94) and that Weinstein et al. (2023) provides a better workflow to handle datasets covering wide geographical areas (L94-108). While this may well be the case, the paragraph includes no citations to previous studies, which is needed before highlighting the research gap. Overall, the introduction would benefit from restructuring and the addition of proper citations.

Minor issues

L55-57: This may be obvious to the authors, but adding citations to previous studies that did this would be nice.

L57-60: Please cite some examples here.

L119: Is "RGB camera mosaic" the correct terminology here? Is this the same as "mosaiced RGB imagery"? The term appeared several times after as well.

L121: What does it mean by 70% accuracy between predicted and hand-annotated tree crowns? Does that mean 70% of the crowns have intersection over union scores over 0.5? Not a must to provide a full explanation to how this figure was derived, but if it could be succinctly summarised in a sentence or two it would be useful to include as this could quite heavily affect dataset usability.

L188-189: Missing comma. Also consider adding a sentence before this to make it easier to understand. Something along the lines of "Many species captured in the original dataset would be lost after data filtering". 

L188-198: Overall, this paragraph is difficult to follow. At the start of the paragraph, the authors were referring to the filtered tree stem dataset collected from the field. It then jumped to "species captured by the model". While I understand that the species in the filtered dataset are the ones included in the species classification models, up to this point, the model has not been mentioned or described yet, which makes it very confusing. I would suggest either (1) making it clear at the very start that you are using the species to build a model, and that you want to evaluate the proportion of species the model captures, or, alternatively, (2) describing the filtered tree stems as "training data" and don't refer to the model until the next paragraph. On a more positive note, the hypothetical example at the end of the paragraph (L195-198) was a nice touch, made things much clearer.

L203-205: What does it mean by optionally classifying trees into "Broadleaf", "Conifer" and "Dominant species class"? Dominant species would either be broadleaved or coniferous. Did you have one model per site? If so, this should be stated at the very start.

L219-223: Were multiple models built using different hyperspectral datasets? A big issue in hyperspectral remote sensing is that models are often not very transferrable across datasets, especially if the imagery was collected in different seasons when the spectral signature of trees would be complicated by phenology.

L223-225: Needs a citation

L245: Is a 40m buffer enough to prevent spatial autocorrelation from affecting accuracy measurements?

L273: 3.56 million trees per site across 24 sites don't give 100,021,471 trees. Or were there trees outside the sites?

L352-354: Citations needed

L356-357: Citations needed

L356-369: The authors made a good point about current studies often being overly focused on accuracy figures over small and neat datasets. I agree that a model that produces reasonable estimates with 70% OA for large areas would be more useful than one that gets 99% in Indiana Pines. It is also true that many studies are trained and validated with manually segmented crowns, which generate accuracy figures that do not account for errors in automatic tree segmentation. However, I have some reservations regarding the authors' claim that the accuracy figures presented here are better at accounting for these errors. The ~30% error associated with generating the DeepForest bounding boxes is arguably not incorporated into the accuracy figures presented in this study. If perfect manually delineated and species-identified crows were used as validation, the classification errors would be substantially higher. I don't think a dataset needs to be perfect to be useful, and the authors might have done the best they could with the dataset available, but it might be a stretch to claim that the accuracy figures presented here are more representative than those in previous work.

Reviewer #2: 

The authors present an interesting study on using deep learning models to create species predictions for individual trees for 24 sites in the National Ecological Observatory Network. Overall I think it is a fine manuscript and I do not have major criticisms. 

I'm happy to see that the output is shared publicly via Google Earth Engine too. 

- A potential downside of combining high resolution imagery data, lidar, and hyperspectral data is that not all of these are available outside the study area, which makes it hard to extend to other regions, and as a result the transferability of the model is difficult. I understand that including the best available data to make the best possible model was the main aim of this project, but it might be interesting to compare the outcome of a model that has, for example, only RGB imagery included. A recent paper in remote sensing (https://www.mdpi.com/2072-4292/15/5/1463) did exactly this and had reasonable accuracy statistics as far as I can tell. 

A similar point for creating a single model per site. I think this is a valid decision, but it would be interesting to see how much worse (if at all?) a single model would perform. 

- The number of training samples for some of the sites, (e.g. SJER, LENO, CLBJ) is very small. Is it realistic to train a model on so few samples? 

- General comment- the site names are mentioned throughout the manuscript but for people not familiar with NEON 

Reviewer #3:

[IMPORTANT: please note this reviewers' additional comments in the attached PDF]

In the manuscript entitled "Individual tree species maps for the National Ecological Observatory Network", the authors introduce a new dataset generated using a rather large dataset using a rather complex methodology. While I think that any effort towards creating open and ML-ready datasets represents a great step in science I have some major concerns regarding this study that I think should be addressed by the authors prior to publication. 

Generally, it seems that the authors aim at generating a very large dataset rather than creating a well-curated dataset. From personal experience and from literature is evident that, if on one hand deep learning models can leverage large datasets, the quality of the labels remains the most important aspect to train models that are reliable and hence do not inherit inherent biases and quality issues caused by having poor training data. This aspect, which due to inherent under detection of trees in RGB images (cannot detect and classify species for understory and partly dominated trees) is clearly an issue in the dataset presented in this paper, and can cause serious problems in downstream applications. In particular, it is rather dangerous when the users are not well aware of the limitations of the dataset. Hence, I urge the authors to clearly highlight the limitation of the proposed dataset, which is composed of pseudo-labelled trees (i.e. predicted from a deep learning models and not manually annotated), artificially generated trees (i.e. trees with 1 m buffer bounding box created according to lines 183-184). This would be best clarified by including a separate sub-section in the discussion section to highlight the dataset key limitations. Here is critical that the authors report and quantify the:

a) under-detection of trees using the bounding box model in terms of detection rate (or estimation of tree numbers) compared to field data. 

b) Under-estimation of species richness by comparing for each site the total number of surveyed tree species in the field (i.e. without thresholds (see specific comment in Table 2).

Overall, as a user I would like to know how many real trees you are correctly identifying and how many are you missing by using a deep learning model that relies on RGB data. 

- The introduction contains a large portion of text (lines 71

---

## [Decision Letter · Decision Letter 2]

25 May 2024

Dear Dr Weinstein,

Thank you for your patience while we considered your revised manuscript entitled "Individual canopy tree species maps for the National Ecological Observatory Network" for publication as a Methods and Resources at PLOS Biology. This revised version of your manuscript has been evaluated by the PLOS Biology editors, the Academic Editor and tow of the original reviewers.

The reviews are attached below. Based on the reviews, we are likely to accept this manuscript for publication, provided you satisfactorily address the remaining points raised by the reviewers. Please also make sure to address the data and other policy-related requests stated below.

We expect to receive your revised manuscript within two weeks. 

*Published Peer Review History*

*Press*

Sincerely,

Ines

--

Ines Alvarez-Garcia, PhD

Senior Editor

PLOS Biology

on behalf of

Roland Roberts, PhD

Senior Editor

PLOS Biology

rroberts@plos.org

DATA POLICY:

Fig. 3; Fig. 4 and Fig. 7

Please also ensure that figure legends in your manuscript include information ON WHERE THE UNDERLYING DATA CAN BE FOUND, and ensure your supplemental data file/s has a legend.

CODE POLICY

Per journal policy, if you have generated any custom code during the curse of this investigation, please make it available without restrictions upon publication. Please ensure that the code is sufficiently well documented and reusable, and that your Data Statement in the Editorial Manager submission system accurately describes where your code can be found. [IF APPLICABLE: As the code that you have generated to XXX is important to support the conclusions of your manuscript, its deposition is required for acceptance.]

Reviewers' comments

Rev. 1: Aland Chan

The manuscript has improved substantially compared to the previous submission. The introduction now includes a better description of previous work, and the revisions largely addressed the concerns raised by reviewers. 

One suggestion for the revised introduction is that it should focus more on the product rather than the methodology. The methodology used in the study has largely been described in Weinstein et al. (2023), so the main novelty of the study is the creation of a large-scale species map. Paragraphs 2-3 in the introduction gave an overview of how methods in tree delineation and species classification developed, which is great, but existing ITC species maps have barely been mentioned. I would suggest modifying paragraph 4 to include some existing studies that successfully produced ITC species maps (e.g. Dalponte et al. (2019) which is also in the NEON network in the US, Lee et al. (2016) in the UK, or others) then highlight how your product cover a broader spatial extent or more forest types. 

It would also be nice to see some discussion on how the product compares to pixel-based products in terms of ecological applications. 

Other than that I have no further comments.

Rev. 3:

Dear authors,

Thank you for thoroughly addressing my comments. I believe you did an excellent job of rephrasing the narrative from "individual trees" to "individual canopy trees," which is a fair compromise and generally improves the clarity of the manuscript.

I also appreciate the inclusion of Figure S3, as it offers a better understanding of your models' actual performance in terms of tree counts. However, I suggest adding the RMSE and mean difference (average of the residuals) as text within each quadrant. This addition would provide a more complete and precise understanding of the random and systematic errors, necessary to address the next comment.

Regarding line 384: Please be more specific than "we expect that …". Based on the mean difference from all datasets in Figure S3, you can estimate the average underestimation. Please rephrase this sentence to provide the mean difference and its relative value (%) compared to the mean number of trees across datasets.

---

## [Editor Report · Decision Letter 3]

5 Jun 2024

Dear Dr Weinstein,

Thank you for the submission of your revised Methods and Resources "Individual canopy tree species maps for the National Ecological Observatory Network" for publication in PLOS Biology. On behalf of my colleagues and the Academic Editor, Andrew Tanentzap, I'm pleased to say that we can in principle accept your manuscript for publication, provided you address any remaining formatting and reporting issues. These will be detailed in an email you should receive within 2-3 business days from our colleagues in the journal operations team; no action is required from you until then. Please note that we will not be able to formally accept your manuscript and schedule it for publication until you have completed any requested changes.

IMPORTANT: I've asked my colleagues to include the following final editorial request among their own (apologies; my colleague omitted to include it in her previous request): "Many thanks for providing the underlying numerical values for Figs 3, 4 and 7. Please could you also provide similar data files for Figs S2 and S3, and cite the names of these files in the corresponding Figure legends?"

Sincerely, 

Roli Roberts

Senior Editor

PLOS Biology

rroberts@plos.org